# A homozygous *FANCM* mutation underlies a familial case of non-syndromic primary ovarian insufficiency

Baptiste Fouquet[1†], Patrycja Pawlikowska[2†], Sandrine Caburet[3], Celine Guigon[4], Marika Mäkinen[5], Laura Tanner[5], Marja Hietala[5], Kaja Urbanska[6], Laura Bellutti[7], Bérangère Legois[3], Bettina Bessieres[8], Alain Gougeon[9], Alexandra Benachi[10], Gabriel Livera[7], Filippo Rosselli[2], Reiner A Veitia[3‡], Micheline Misrahi[1‡]*

[1]Faculté de Médecine, Université Paris Sud, Université Paris Saclay, Hôpital Bicêtre, Le Kremlin-Bicêtre, France; [2]CNRS UMR8200,Equipe labellisée La Ligue Contre Le Cancer, Université Paris Sud, Université Paris Saclay, Gustave Roussy, Vilejuif, France; [3]Institut Jacques Monod, Université Paris Diderot, Paris, France; [4]Université Paris-Diderot, CNRS, UMR 8251, INSERM, U1133, Paris, France; [5]Department of Clinical Genetics, Turku University Hospital, Turku, Finland; [6]CNRS UMR8200, Université Paris Sud, Université Paris Saclay, Villejuif, France; [7]UMR967 INSERM, CEA/DRF/iRCM/SCSR/LDG, Université Paris Diderot, Sorbonne Paris Cité, Université Paris-Sud, Université Paris-Saclay, Fontenay aux Roses, France; [8]Department of Histology, Embryology and Cytogenetics, Hôpital Necker-enfants malades, Paris, France; [9]UMR Inserm 1052, CNRS 5286, Faculté de Médecine Laennec, Lyon, France; [10]Department of Obstetrics and Gynaecology, AP-HP, Université Paris-Sud, Université Paris-Saclay, Clamart, France

*For correspondence:
Micheline.misrahi@aphp.fr

†These authors contributed equally to this work
‡These authors also contributed equally to this work

Competing interests: The authors declare that no competing interests exist.

**Abstract** Primary Ovarian Insufficiency (POI) affects ~1% of women under forty. Exome sequencing of two Finnish sisters with non-syndromic POI revealed a homozygous mutation in *FANCM,* leading to a truncated protein (p.Gln1701*). *FANCM* is a DNA-damage response gene whose heterozygous mutations predispose to breast cancer. Compared to the mother's cells, the patients' lymphocytes displayed higher levels of basal and mitomycin C (MMC)-induced chromosomal abnormalities. Their lymphoblasts were hypersensitive to MMC and MMC-induced monoubiquitination of FANCD2 was impaired. Genetic complementation of patient's cells with wild-type FANCM improved their resistance to MMC re-establishing FANCD2 monoubiquitination. *FANCM* was more strongly expressed in human fetal germ cells than in somatic cells. FANCM protein was preferentially expressed along the chromosomes in pachytene cells, which undergo meiotic recombination. This mutation may provoke meiotic defects leading to a depleted follicular stock, as in *Fancm*-/- mice. Our findings document the first Mendelian phenotype due to a biallelic *FANCM* mutation.

DOI: https://doi.org/10.7554/eLife.30490.001

## Introduction

Primary Ovarian Insufficiency (POI) affects about 1% of women under forty years. It is often diagnosed too late, thus generating infertility and significant morbidity and mortality due to steroid-deprivation associated symptoms. Infertility is usually definitive but resumption of ovarian function occurs in ~24% of cases (*Tucker et al., 2016*). POI is etiologically heterogeneous (OMIM: ODG1 # 233300. ODG2 # 300510 ODG3 # 614324, ODG4 # 616185) and remains idiopathic in ~70% of the

**eLife digest** About one in 100 women under the age of 40 experience a condition known as primary ovarian insufficiency, which is sometimes known as premature menopause. Women with this condition may have fewer egg cells and are usually infertile. Women with primary ovarian insufficiency are also more at risk of other diseases, including the bone disorder osteoporosis and cardiovascular diseases. The condition is thought to have a genetic basis in part, although so far its causes are largely unknown.

Fouquet, Pawlikowska et al. looked at all the genes in genomes of three women in one Finnish family: two sisters and their mother. Both of the sisters had primary ovarian insufficiency, but were otherwise healthy. Their mother did not have the condition. The genetic analysis identified a mutation in a gene called *FANCM*, which is involved in the cell's repair response to DNA damage and has recently been linked to breast cancer. This gene is mostly active in egg cells within the ovary. The sisters' protein made from this mutated copy of the gene was cut short compared with the protein produced by the mother's *FANCM* gene.

Due to the mutation, the sisters were more sensitive to chemicals that can damage the DNA, effectively making their genome less stable. The affected sisters also had higher levels of abnormalities in the chromosomes compared with their unaffected mother. Fouquet, Pawlikowska et al. then inserted a healthy version of the *FANCM* gene into the sisters' cells. This reversed the sensitivity of the sisters' cells to DNA-damaging chemicals.

The findings confirm a genetic link between primary ovarian insufficiency and genes responsible for DNA repair. Mutations in these genes can also make people more at risk of certain cancers. The findings point towards offering some women who have primary ovarian insufficiency in-depth genetic counselling with a long-term follow-up, when alterations of cancer-susceptibility genes are responsible for their condition.

DOI: https://doi.org/10.7554/eLife.30490.002

cases, but a number of genetic variants have been identified, including mutations in meiotic and DNA repair genes (*Tucker et al., 2016*). Consistently, several DNA repair and genomic instability disorders, such as Fanconi anemia (FA), are known to be associated with hypogonadism, ovarian failure and/or infertility. FA is a bone marrow failure syndrome accompanied by developmental defects, predisposition to leukemia, chromosome fragility and hypersensitivity to DNA interstrand crosslinks (ICL) (*Bogliolo and Surrallés, 2015*; *Ceccaldi et al., 2016*). The products of the 21 genes (FANCA to FANCV) whose loss-of-function has been associated to FA are subdivided into three functional groups (*Bogliolo and Surrallés, 2015*; *Ceccaldi et al., 2016*; *Wang, 2007*). The first one, the FANC-core complex consisting of seven FA proteins (A, B, C, E, F, G, and L) and two FA-associated proteins (FAAP20 and FAAP100), is assembled in response to DNA damage and/or stalled replication forks to monoubiquitinate FANCD2 and FANCI (the second group). This monoubiquitination allows the FANCD2-FANCI heterodimer to coordinate the DNA repair/replication rescue activities of the third group of FANC proteins, which includes nucleases and proteins involved in homologous recombination. The third group includes BRCA1, BRCA2, RAD51, PALPB2 and BRIP1, whose mutations predispose to breast and ovarian cancer (BOC) (*Bogliolo and Surrallés, 2015*). However, a more recent analysis reconsidered the role of some FA-associated genes in the establishment of *bona fide* FA clinical and cellular phenotypes and excluded *FANCM* from the group of FA genes (*Bogliolo and Surrallés, 2015*). The phenotypes associated to FANCM biallelic mutations thus far are cancer predisposition, in particular early-onset breast cancer in females, and chemosensitivity. (*Michl et al., 2016*; *Bogliolo et al., 2017*; *Catucci et al., 2017*).

Here, we have performed a whole-exome sequencing in a Finnish family with two patients presenting with non-syndromic POI and identified a homozygous truncating mutation (c.5101C>T; p. Gln1701*) in *FANCM*, which explains POI in these patients.

## Results

The two POI patients belong to a consanguineous family (*Figure 1A*) without history of cancer. The parents and the 20 year old brother are reported as healthy, and the mother had regular menses at

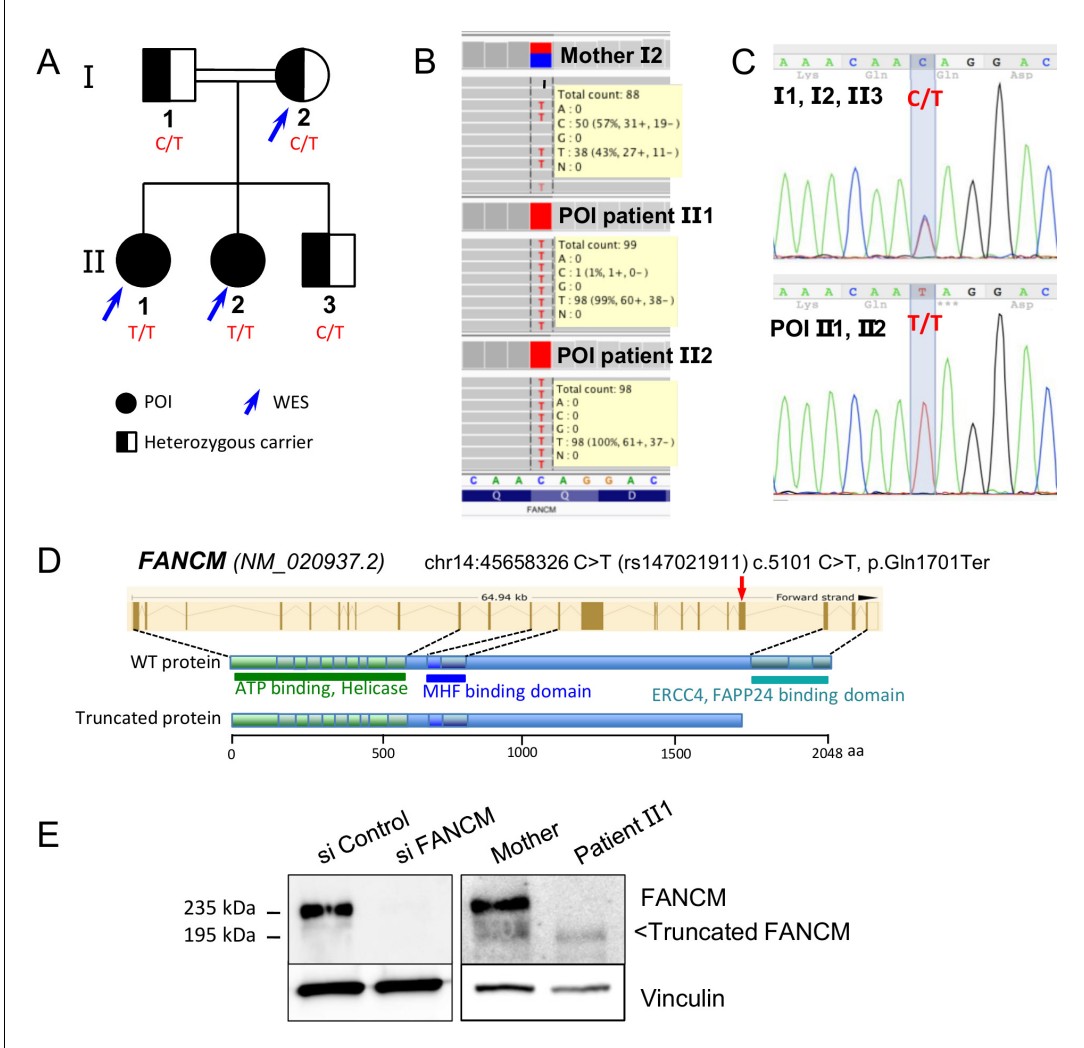

**Figure 1.** Molecular analysis of the POI family. (**A**) Pedigree of the Finnish family with two sisters affected with POI. (**B**) Exome data visualization in IGV (Integrative Genomics Viewer) shows a high coverage of the variant position and a large number of reads in the three individuals. (**C**) Sanger sequencing confirmed the presence of the variant at a homozygous state in both affected sisters and at a heterozygous state in both parents and their brother. (**D**) Structure of the *FANCM* gene and protein, and position of the causal variant. (**E**) Western blots of FANCM in the POI family. HEK293 cell transfected with a FANCM-specific siRNA were used to validate the specificity of the anti-FANCM antibody. To transiently deplete FANCM, HEK293 cells were transfected with 20 nmol/L of small interfering RNA (siRNA) targeting FANCM, 5′-GGC-UAC-GUC-CAG-GAG-CGC-3′ with the CaCL$_2$ method. Panel present the result of a representative experiment on at least three independent analysis.

DOI: https://doi.org/10.7554/eLife.30490.003

47 years. Both patients had normal pilosity, breast and external genitalia, normal blood counts and liver balance, and had normal high-resolution karyotype and *FMR1* gene. No abnormalities in skin pigmentation and skeletal development were observed.

Proband had menarche at 12 years of age, with irregular cycles (20–60 days). Hormonal contraception was started at the age of 16 years for menorrhagia and stopped four years later, after which menstrual cycles became highly irregular (21–140 days). At 24 years, she had hot flushes for about one year and spaniomenorrhea. Blood hormonal assays and ultrasonographic studies of the ovaries are reported in *Table 1*. FSH level was high (41 IU/l) and AMH low. POI was diagnosed. At the age of 26, hormonal stimulation was attempted with no response. Soon after that, FSH increased to 120 IU/l and hormonal replacement was initiated.

Her sister had menarche at 12 years of age, with regular menses (23 days). Hot flushes started at the age of 20 years. At 22, she consulted for hot flushes and spaniomenorrhea. She had elevated

**Table 1.** Clinical, hormonal and ultrasonography studies of patients with *FANCM* mutation

| Case | Menstrual cycles | Age at evaluation | BMI (Height cm/ weight Kg) | FSH IU/l | LH IU/l | E2 nmol/l | AMH ng/ml | T nmol/l | PRL mU/l | TSH mU/l | Ovarian surface (R/L) mm | Presence of follicles at US |
|---|---|---|---|---|---|---|---|---|---|---|---|---|
| Patient-1 | Secondary amenorrhea (24 years) | 24 | 22,27 (160/57) | 41 | 38 | 0.388 | | 1.8 | 254 | 2.6 | 20 × 17 cyst/ 16 × 8 | Yes (2) |
| | | 26 | | 120 | | 0.053 | <0.2 | | 285 | | 26 × 11/ 18 × 8 | Yes (1) |
| | | 28 | | 65 | 51 | 0.120 | <0.2 | 0.96 | | | | |
| Patient-2 | Spanio menorrhea (22 years) | 22 | 24,46 (163/65) | 16 | 4 | 0.069 | | | 843 | | | |
| | | 23 | | 16 | | | 0.6 | | NA | 1.7 | 18 × 11/ 19 × 10 | No |

BMI: body mass index US: ultrasonography; E2: estradiol; T: testosterone; P: progesterone; PRL: prolactin; NA: not available
Follicular phase: 3.5-12.5 [FSH (IU/l)], 2.4-12.6 [LH (IU/l)], 0.11-0.22 [E2 (nmol/l)]
Ovulatory phase: 4.7-21.5 [FSH (IU/l)], 14-96 [LH (IU/l)], 0.42-1.40 [E2 (nmol/l)]
Luteal phase 1.7-7.7 [FSH (IU/l)], 1-11.4 [LH (IU/l)], 0.17-0.79 [E2 (nmol/l)]
Menopause 26-135 [FSH (IU/l)], 8-33 [LH (IU/l)], $\leq$ 0.05 [E2 (nmol/l)]

DOI: https://doi.org/10.7554/eLife.30490.004

FSH (16 IU/l) and low E2 levels (*Table 1*). Incipient POI was diagnosed. As prolactin was elevated and brain MRI showed a suspected 3 mm adenoma, treatment with Parlodel was initiated. However, FSH remained elevated and AMH low (*Table 1*). At 23, hormonal stimulation was initiated with poor results. However, about 6 months later, she became spontaneously pregnant and gave birth to a healthy child.

## Identification of a FANCM mutation by whole-exome sequencing and molecular studies

Both sisters with POI and their mother were studied by whole-exome sequencing, which generated ~45 millions of mapped read pairs per sample (91% of targeted exome at $\geq$10X). Exome capture, sequencing and data processing were performed as described in (*Fauchereau et al., 2016*). Variants were filtered according to the following criteria: minimum depth of 5 reads and variant quality of 20, with potential impact on transcript or protein, homozygosity in both affected sisters and heterozygosity in the mother and minor allele frequency (MAF) under 1%. The only variant fulfilling these criteria was the non-sense mutation chr14:45658326 C/T (rs147021911) in exon 20 of FANCM (*Figure 1B*, *Tables 2–3*). According to the ExAC database, its MAF is 0.0013, slightly higher in the Finnish population (0.0089), with only one homozygous individual described in the ExAC

**Table 2.** Whole Exome sequencing metrics in the Finnish family with FANCM Q1701* variant.

| WES metrics | mother | POI 1 | POI 2 |
|---|---|---|---|
| Gbases | 6.682 | 7.273 | 6.931 |
| Number of reads (millions) | 44.5 | 48.5 | 46.2 |
| % Alignment | 97.03 | 97.12 | 97.09 |
| % Mismatch Rate R1 | 0.19 | 0.19 | 0.2 |
| % Mismatch Rate R2 | 0.31 | 0.31 | 0.32 |
| %$\geq$Q30 bases | 96.34 | 96.26 | 96.15 |
| Mean Quality Score | 39.4 | 39.4 | 39.4 |
| Mean Depth (X) | 68 | 74 | 70 |
| % of bases covered at 25X | 83 | 84 | 83 |

DOI: https://doi.org/10.7554/eLife.30490.005

**Table 3.** Whole Exome sequencing Variant filtering in the Finnish family with FANCM Q1701* variant

| Variants called in | Mother | POI 1 | POI 2 |
|---|---|---|---|
| Total | 40621 | 40343 | 40561 |
| SNPs | 37929 | 37590 | 37859 |
| Indels | 2692 | 2753 | 2702 |

| Variant filters | # of variants |
|---|---|
| Shared between POI 1 and POI 2 | 32784 |
| in coding sequence or splice | 17207 |
| in CDS, but not synonymous | 8855 |
| homozygous in both POI 1 and POI 2 | 4053 |
| and heterozygous in Mother | 512 |
| MAF < 1% in EVS and 1000G and ExAC | 5 |
| <1% in IG exomes | 2 |
| with high potential functional impact | 1 |

DOI: https://doi.org/10.7554/eLife.30490.006

database. The variant and its segregation in the family were verified by Sanger sequencing of the exon 20 of the *FANCM* (*Figure 1C*). This variant leads to the p.Gln1701* truncation at the protein level, which removes the C-terminal endonuclease and the FA associated protein 24 (FAAP24)-interaction domain (*Figure 1D*). FAAP24 mediates chromatin association of FANCM.

EBV-immortalized cells were obtained from patient 2 and the mother. Immunoblot with an anti-FANCM antibody confirmed the expression of a truncated FANCM protein of ~195 kDa in the cell line derived from the patient, whereas both the wild-type/WT (MW ~235 kDa) and the truncated proteins were present in the heterozygous mother (*Figure 1E*). The expression level of the truncated protein was significantly reduced compared to the WT, suggesting that the mutation destabilizes either the protein or the corresponding mRNA (i.e., non-sense mediated decay).

No variants were observed in other FANC genes. Indeed, we have exhaustively looked for the presence of variants in a list of FA genes (*Table 4*). The very good coverage of each gene (80-90X in the POI patients) excludes the possibility of having missed a variant due to a lack of coverage. The presence of heterozygous variants, both in coding regions and elsewhere (UTRs, the covered intronic sequences, etc.), in one or both POI patients, excludes the possibility of hemizygosity due to heterozygous deletions. The ratios of the number of reads of the alleles for these heterozygous variants were unbiased (i.e., close to 1) and argue against gene deletions. The detection of variants outside the coding portions of the exons also argues against the possibility of partial deletions. Exome-based CNV analysis showed that the detected CNVs were unrelated to the disease as they were either not common to the two sisters or were present in the mother. Finally, the total lack of correctly segregating variants (i.e common to both POI patients and heterozygous in the mother) in FA genes excludes the possibility of their involvement in the pathology.

## FANCM is expressed in germ cells of human fetal and adult ovaries

FANCM expression was studied in germ cells of human fetal and adult ovaries. Human fetal material was obtained from the Antoine Béclère Hospital (Clamart, France) following legally-induced abortions or therapeutic termination of pregnancy. The identification of meiotic stages was performed in histological pieces that we have previously characterized (*Frydman et al., 2017*) and based on the chromatin features. qRT-PCR in human fetal ovaries demonstrated that FANCM mRNAs were expressed throughout ovarian development (5–32 weeks post-fertilization, wpf) (*Figure 2A*). Of note, expression tended to be higher than average in 14 and 17 wpf ovaries that are stages containing the highest proportion of germ cells progressing into meiotic prophase I.

Cell-sorting experiments conducted in 8–12 wpf ovaries indicated that FANCM transcripts were predominant in oogonial cells (D2-40-positive) compared to somatic cells (*Figure 2B*). Immunohistochemical studies in human fetal ovaries show that FANCM protein was present in the nuclei of oogonia but staining was stronger in pachytene stage oocytes (i.e., at 8 and 14 wpf respectively in

**Table 4.** Compilation of Whole-Exome-Sequencing data for the genes of the FANC pathway, excluding all the genes except FANCM as potentially harboring a causative variant.

| Gene name | Alias | Mean depth in targeted exons (1) | | | Nb variants in | Htz variants | Mean ratio for htz allelic | Presence of htz variants (5) | | | | | Nbr of rare variants (6) | |
|---|---|---|---|---|---|---|---|---|---|---|---|---|---|---|
| | | mother | POI 1 | POI 2 | family (2) | in >= 1 sister (3) | reads in sisters (4) | upstream | 5'UTR | deep intronic | 3'UTR | downstream | correct segregation | and pathogenic |
| FANCA | | 87.9 | 94.9 | 90.3 | 29 | yes | 0.92 | no | no | yes | yes | yes | 0 | 0 |
| FANCB | | 88.6 | 93.8 | 89.9 | 2 | yes | 1.14 | no | no | no | no | no | 0 | 0 |
| FANCC | | 64.4 | 71.5 | 68.2 | 0 in this family | | | | | | | | | |
| FANCD1 | BRCA2 | 84.6 | 90.5 | 84.5 | 12 | yes | 1.01 | yes | yes | no | no | no | 0 | 0 |
| FANCD2 | | 76.1 | 82.1 | 75.0 | 2 | yes | 0.825 | no | no | no | yes | yes | 0 | 0 |
| FANCE | | 89.5 | 100.2 | 99.0 | 2 | yes | 1.08 | no | no | no | no | no | 0 | 0 |
| FANCF | | 143.0 | 150.2 | 139.7 | 0 in this family | | | | | | | | | |
| FANCG | | 123.5 | 139.8 | 130.2 | 1 | yes | 1.22 | no | no | no | no | no | 0 | 0 |
| FANCI | | 75.7 | 80.3 | 76.6 | 9 | yes | 0.89 | no | no | no | no | yes | 0 | 0 |
| FANCJ | BRIP1 | 104.2 | 113.0 | 107.7 | 4 | yes | 0.76 | no | no | yes | no | no | 0 | 0 |
| FANCL | | 80.2 | 88.5 | 80.7 | 3 | yes | 0.96 | no | no | no | no | no | 0 | 0 |
| FANCM | | 86.9 | 91.3 | 87.8 | 1 | no | / | no | no | no | no | no | 1 | 1 |
| FANCN | PALB2 | 97.0 | 106.9 | 101.1 | 0 in this family | | | | | | | | | |
| FANCO | RAD51C | 92.1 | 103.2 | 94.9 | 1 | yes | 0.77 | no | yes | no | no | no | 0 | 0 |
| FANCP | SLX4 | 107.9 | 116.4 | 112.8 | 15 | yes | 0.88 | no | no | no | yes | no | 0 | 0 |
| FANCQ | ERCC4 | 91.5 | 98.1 | 90.9 | 2 | yes | 1.21 | no | no | yes | no | no | 0 | 0 |
| FANCR | RAD51 | 75.9 | 82.4 | 74.5 | 0 in this family | | | | | | | | | |
| FANCS | BRCA1 | 87.6 | 94.3 | 86.8 | 1 | yes | 0.85 | no | no | no | no | no | 0 | 0 |
| FANCT | UBE2T | 53.9 | 63.2 | 52.6 | 2 | yes | 0.91 | no | no | no | no | no | 0 | 0 |
| FANCU | XRCC2 | 104.3 | 116.1 | 113.1 | 0 in this family | | | | | | | | | |
| FANCV | MAD2L2 | 72.6 | 85.4 | 74.3 | 0 in this family | | | | | | | | | |
| FAAP100 | C17orf70 | 69.0 | 66.8 | 69.1 | 7 | yes | 0.94 | no | yes | yes | yes | no | 0 | 0 |
| FAAP24 | C19orf40 | 58.3 | 64.6 | 59.2 | 2 | yes | 0.89 | no | no | no | no | no | 0 | 0 |
| FAAP20 | C1orf86 | 57.7 | 62.0 | 59.7 | 3 | yes | 1.20 | yes | no | yes | yes | no | 0 | 0 |
| FAAP16 | APITD1 | 37.7 | 44.2 | 40.7 | 4 | yes | 1.12 | no | yes | yes | no | no | 0 | 0 |
| FAAP10 | STRA13 | 57.3 | 62.1 | 57.8 | 4 | yes | 1.18 | no | yes | no | no | yes | 0 | 0 |
| FAN1 | | 101.8 | 112.9 | 102.6 | 4 | yes | 0.97 | no | no | no | yes | no | 0 | 0 |
| | mean | 84.0 | 91.7 | 85.9 | 110 | | 0.99 | | | | | | | |

(1) For each gene, the mean depth per exon was averaged over all exons of the gene. The high coverage for all genes excludes the possibility of not detecting a causative variant in other FANC genes.

(2) Total number of upstream, downstream, 5' and 3' UTRs, intronic, synonymous, splice site, missense, frameshift and stop variants in each gene.

(3) The presence of heterozygous variants in at least one of the patients excludes the possibility of hemizygosity for all genes (with the exception of the variant in FANCM that is homozygous)

(4) The ratio between the number of reads for each allele was averaged for all heterozygous variants in the two affected sisters. A ratio close to one for each gene indicates no bias and argues against a possible deletion for the gene.

(5) The presence of heterozygous variants in the various genic portions argues against the possibility of partial deletions.

(6) Among the 110 variants detected in the genes included in the FANC pathway, only the non-sense variant found in FANCM is homozygous in both patients, is rare (below 1% in ExAC database) and predicted as pathogenic.
DOI: https://doi.org/10.7554/eLife.30490.007

*Figure 2C*). Of note, staining localized along the chromosomes in pachytene cells that undergo meiotic recombination. FANCM was also observed in oocytes arrested at the diplotene stage of prophase I during the last trimester of pregnancy and in adults (*Figure 2C*). The co-staining with Synaptonemal complex protein 3 (SYCP3) or DEAD box protein 4 (DDX4/VASA) confirmed respectively the meiotic and germinal nature of the FANCM-positive cells (*Figure 2D*).

### The *FANCM* mutation leads to hypersensitivity to Mitomycin C and altered FANCD2 monoubiquitination in response to DNA interstrand crosslinks but not to replication inhibition

Next, we monitored chromosome breakage in primary lymphocytes from the two patients and their mother. Baseline and DNA-damage induced chromosome breakage and rearrangement were blindly scored on 50 metaphases without treatment or after exposure to Mitomycin C (MMC) for 72 hr. In line with a role of FANCM in the maintenance of genome stability, the occurrence of chromosome breakages and rearrangements was higher in both POI patients than in their mother (*Figure 3A and B*).

We also determined the impact of the *FANCM* truncating mutation on the sensitivity to Mitomycin C and on FANCD2 monoubiquitination in response to DNA interstrand crosslinks and replication inhibition. In response to MMC, primary lymphocytes from both patients had a reduced capability to monoubiquitinate FANCD2 (*Figure 3C*). The reduced level of FANCD2 in lymphocytes from POI patients is likely due to a reduced proliferation of their cells in culture conditions. Next, we determined the growth inhibition response of the lymphoblasts from POI patient-2 and her mother in response to MMC (*Figure 3D*), as previously described (*Ridet et al., 1997*). The cells from the mother behaved like FANC-pathway proficient cells, while the response of the mutated cells was similar to that of FANCA- and FANCC-deficient lymphoblasts.

Finally, we assessed FANCD2 monoubiquitination in lymphoblastoid cells in response to MMC, that arrests replication by inducing DNA lesions (*Figure 4A*), or to two replication inhibitors, hydroxyurea (HU) and aphidicolin (APH), that block replication forks by poisoning DNA polymerases (*Figure 4B*). Surprisingly, whereas in response to MMC, FANCD2 monoubiquitination was clearly impaired in FANCMmut cells, those cells maintained a residual but detectable capability to monoubiquitinate FANCD2 after HU or APH treatments. Finally, consistent with a proficient DNA damage and stalled replication forks signaling in FANCM mutated cells, we failed to observe any major impairment in the MMC- phosphorylation of H2AX and CHK1 (*Figure 4A*) (*Durocher and Jackson, 2001*).

To further validate that the identified bi-allelic mutation in *FANCM* was responsible for the MMC hypersensitivity observed in the patient's lymphoblasts, we transduced them with a FANCM-WT cDNA-expressing lentiviral vector (see Materials and methods). Transiently genetically complemented cells recovered a significant resistance to MMC (*Figure 3D*) as well as an improved monoubiquitination of FANCD2 in response to MMC (*Figure 3E*).

## Discussion

The two patients with POI studied here belong to a consanguineous family and are thus homozygous for the FANCM mutation inherited from their parents. Our *FANCM* mRNA and protein expression studies in the developing human ovary suggest that the expression of *FANCM* starts in mitotic germ cells (first trimester of pregnancy) notably along chromosome axes and increases at the onset of meiosis (second trimester) that are maintained up to the diplotene stage in follicles. Given the known role of FANCM to sustain primordial germ cell proliferation in mouse (*Luo et al., 2014*) and its conserved function during meiotic recombination (*Lorenz et al., 2012*; *Crismani et al., 2012*), it is likely that both processes are sensitive to a *FANCM* mutation.

FANCM is a nuclear partner of the FANCcore complex, belonging to the FANC pathway that promotes DNA repair and safeguards replication. Indeed, FANCM is targeted to damaged DNA and/or

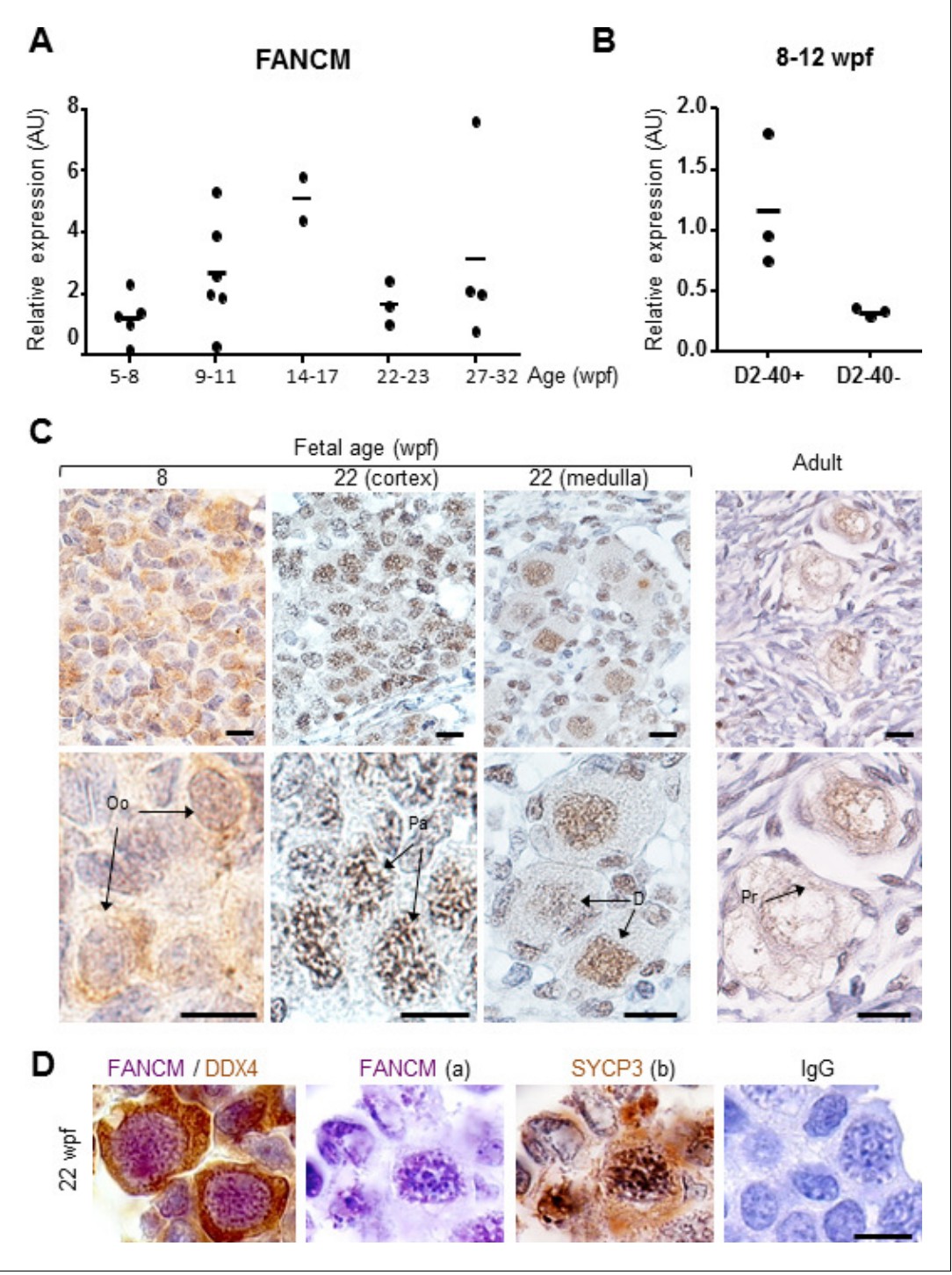

**Figure 2.** *FANCM* expression in human fetal ovaries. (**A**) Relative *FANCM* mRNA abundance was measured by RT-qPCR in human fetal ovaries from 5 to 32 weeks post-fertilization (wpf). (**B**) Germ cells (D2-40+) and somatic cells (D2-40-) were sorted from three ovaries ranging from 8 to 12 wpf and *FANCM* expression was measured. *ACTB* was used to normalize *FANCM* expression in all samples. Dots represent different ovaries and the mean is indicated by the line. (**C**) Immunohistochemistry of FANCM in human fetal and adult ovaries. Fetal ovaries at 8 and 22 wpf and adult ovaries were studied. FANCM positive cells appear in yellow/brown color (monoclonal FANCM CV5.1 antibody, Novus Biologicals, Abingdon, UK). Ovarian sections were counterstained with hematoxylin (blue staining). Oo, oogonia; Pa, oocyte at the pachytene stage of meiosis I, D, oocyte at the diplotene stage of meiosis I; Pr, oocyte in primordial follicle. (**D**) Co-staining in 22 wpf ovaries, for FANCM (purple) and DDX4 (brown)

*Figure 2 continued on next page*

*Figure 2 continued*
confirmed the germ cell identity of FANCM-positive cells (left). Successive staining for FANCM and SYCP3 in the same section (panels a and b). Negative control performed with non-immune mouse IgG (right). Scale bar: 10 µm.
DOI: https://doi.org/10.7554/eLife.30490.008

stalled replication forks by its partners FAAP24, FANCM-interacting histone fold protein 1 (MHF1) and 2 (MHF2), where it recruits the FANCcore complex to monoubiquitinate FANCD2 and FANCI and coordinates DNA repair/replication rescue (*Ceccaldi et al., 2016*; *Michl et al., 2016*) allowing the faithful transmission of undamaged chromosome to the daughter cells. Consistently, the occurrence of chromosome breakages and rearrangements was higher in both POI patients than in their mother (*Figure 3A and B*). Our biochemical studies show that lymphocytes from both patients have a reduced capability to monoubiquitinate FANCD2 in response to MMC pointing to a defect in the activation of the FANCcore complex in FANCMmut cells. However the cells maintained a detectable capability to monoubiquitinate FANCD2 in response to HU or APH treatments.

A previous study links POI to other breast cancer genes such as BRCA1 (*Oktay et al., 2010*). Although recent works established a link between the heterozygous c.5101C>T *FANCM* truncating mutation and BOC predisposition in the Finnish population (*Kiiski et al., 2016*, *2014*; *Neidhardt et al., 2017*), there is no history of BOC in the family investigated here. Interestingly, a few apparently healthy (i.e., without cancer) individuals homozygous for the c.5101C>T FANCM mutation were identified in previous studies (*Kiiski et al., 2016*, *2014*; *Neidhardt et al., 2017*). Indeed, the authors stated, by direct observation or by looking at registered clinical data, that none of them presented FA stigmata (although no data on age, sex or clinical findings was made available). This is in line with the fact that previously identified bi-allelic inactivating *FANCM* mutations in a few FA patients co-occurred with mutations in other FANC genes, which were indeed responsible for FA (*Chang et al., 2014*; *Singh et al., 2009*). This has led to the recent exclusion of *FANCM* from the group of the bona fide FA genes (*Bogliolo and Surrallés, 2015*). Very recently two studies report that the phenotypes associated to FANCM biallelic mutations thus far are cancer predisposition, in particular early-onset breast cancer in females, and chemosensitivity (*Catucci et al., 2017*; *Bogliolo et al., 2017*). In one study (*Catucci et al., 2017*), two female patients out of five had premature menopause and one patient had an early reduction of the ovarian reserve according to her AMH levels.

Our results suggest that the cells that harbor biallelic c.5101C>T *FANCM* mutations maintain a FANCM residual activity allowing them to cope with the stalling of the replication forks during their normal proliferation in vivo, probably protecting individuals from the hematopoietic and developmental abnormalities that constitute the *bona fide* features of FA. However, the repair defects associated to the c.5101C>T truncating mutation that we describe likely leads to meiotic defects and oocyte apoptosis. Oocytes deficient for DNA repair may accumulate double-strand DNA breaks over time, resulting in reduced oocyte viability (*Oktay et al., 2015*). Along similar lines, in *Fancm*$^{-/-}$ female mice, the ovarian cortex is depleted of primary follicles and the number of developing follicles is reduced compared to WT ovaries. However, due to a bias against *Fancm*$^{-/-}$ females, their fertility was not thoroughly investigated (*Bakker et al., 2009*). Antral follicles were present, albeit in reduced number, which is consistent with the observations in our patients. Some follicles might escape the DNA repair defects and achieve maturation, as epitomized by the presence of antral follicles and corpora lutea in *Fancm*-deficient mice and by the spontaneous resumption of ovarian function with pregnancy reported in one of our patients. In conclusion, in this report, we document the first case implicating *FANCM* mutations in non-syndromic POI. Our findings clearly support a genetic link between infertility and DNA-repair/cancer genes and show the necessity to perform an enhanced genetic counseling of POI patients with a long-term follow-up.

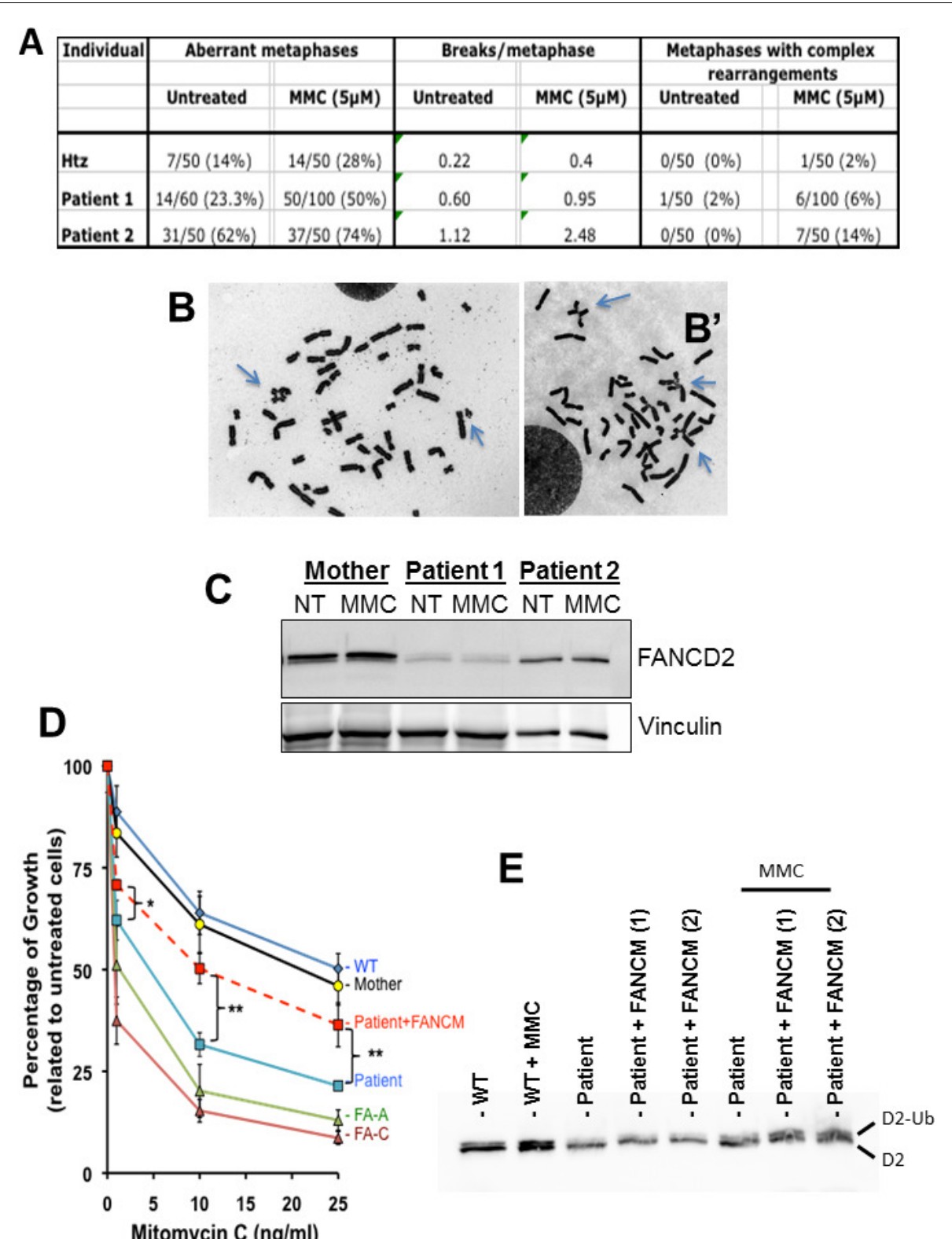

**Figure 3.** Chromosomal breakage and hypersensitivity to Mitomycin C associated to the *FANCM* 5101C > T mutation. (A) Spontaneous and mitomycin C-induced chromosome breakage in primary lymphocytes from the mother and the two POI sisters. Data presented in table are the result of a single experiment. A minimum of 50 metaphases were scored for each sample. (B and B') Examples of untreated and MMC-treated metaphases from patient-2. Arrows indicate breaks and chromosome rearrangements, that is radial figures. (C) Western blot showing FANCD2 expression and

*Figure 3 continued on next page*

*Figure 3 continued*

monoubiquitination in the same lymphocytes than in A and B. Immunoblot analysis were performed using mouse monoclonal anti-FANCM antibody (CV5.1), mouse monoclonal anti-FANCD2, (Santa-Cruz Biotechnology, Dallas, Texas, USA), rabbit anti-FANCA (Abcam, Cambridge, UK), mouse anti-vinculin (Abcam). Panel present the result of a representative experiment on at least three independent analysis. (D) MMC-induced growth inhibition in lymphoblasts from POI patient-2 and her mother compared to the response of cells from a FANCA (HSC-72), a FANCC (HSC-536), a healthy donor (HSC-93, WT) as well as to the patient cells transduced with a WT FANCM-cDNA. The points on the lines represent the means of 3 to 7 independent experiments ± S.D. *p<0,05, **p<0.01, Unpaired Student's T test. (E) The recovery of the MMC-induced FANCD2-monoubiquitination following FANCM expression in cells from patient 2 is shown. (1) and (2) indicate two independent experiments. Cells were treated with MMC (500 ng/ml) and proteins extracted 24 hr later.

DOI: https://doi.org/10.7554/eLife.30490.009

The following source data is available for figure 3:

**Source data 1.** File presents the original data of each growth inhibition experiment used to calculate mean and S.D. for *Figure 3D*.
DOI: https://doi.org/10.7554/eLife.30490.010

## Materials and methods

### Ethics statement

The study was approved by all the institutions involved. All participants gave informed consent for the study and the study was approved by the agence de Biomedecine (reference number PFS12-002).

### Study approval

Written informed consent was received from participants prior to inclusion in the study and the institutions involved.

### Hormonal assays

FSH, LH, Prolactin, TSH: Electrochemiluminescence immunoassay ECLIA, cobas kit insert Elecsys and cobas e analyzers (2013–10, V 19), Roche Diagnostics GmbH/Roche Diagnostics GmbH, Sandhofer Strasse 116, D-68305 Mannheim, Germany. Equipment: cobas 8000 e 602, Roche Diagnostics. Estradiol: radioimmunological assay, Spectra kits $^{125}$I Coated Tube Radioimmunoassay kit insert, Orion Diagnostica, Espoo, Finland.. Equipment: Gammamaster 1270, Wallac Oy, Turku, Finland. AMH: AMH Gen II ELISA. Equipment: Beckman Coulter, Inc. 250 s. Kraemer Blvd. Brea, CA 92821 U.S.A.

### Cell lines and infection with a lentiviral vector

Blood samples were collected for hormonal and genetic studies and for the production of EBV-immortalized lymphoblastoid cell lines, which was performed at the Genopole, Evry, France, following a standard in-house protocol. EBV-immortalized cells were obtained for patient 2 and her mother. HEK293 (Research Resource Identifier/RRID:CVCL_0045) cells and FANC pathway-proficient (HSC-93 (RRID:CVCL_G049)), and -deficient (HSC-72 (RRID:CVCL_G047), HSC-536 (RRID:CVCL_G045) and GM16756 (RRID:CVCL_G041)) cells have been previously described (*Bourseguin et al., 2016*). HSC-72, HSC-536 and GM16756 bring bi-allelic inactivating mutations in *FANCA*, *FANCC* and *FANCD2*, respectively. Lymphoblastoid cells were cultured in RPMI 1640 medium, supplemented with 12% FCS and antibiotics. Wild-type FANCM cDNA-expressing lentiviral vectors were a gift of M. Bogliolo (Dpt of Genetics and Microbiology, Universitat Autonoma de Barcelona, Spain). Production and titration of lentiviral particles were performed as described (*Hamelin et al., 2006*). The infection was performed on retronectin-coated plates (TaKaRa Bio, CA, USA) and efficiency was assayed by testing GFP expression using flow cytometry.

### Whole-exome and Sanger sequencing

Library preparation, exome capture, sequencing and data processing were performed by IntegraGen SA (Evry, France) according to their in-house procedures. Data analysis was performed as described in (*Fauchereau et al., 2016*). Briefly, genomic DNA libraries were prepared from 600 ng of genomic DNA from three individuals (the mother and the two affected sisters) with NEBNext Ultra kit (New England Biolabs). Target capture, enrichment and elution were performed according to

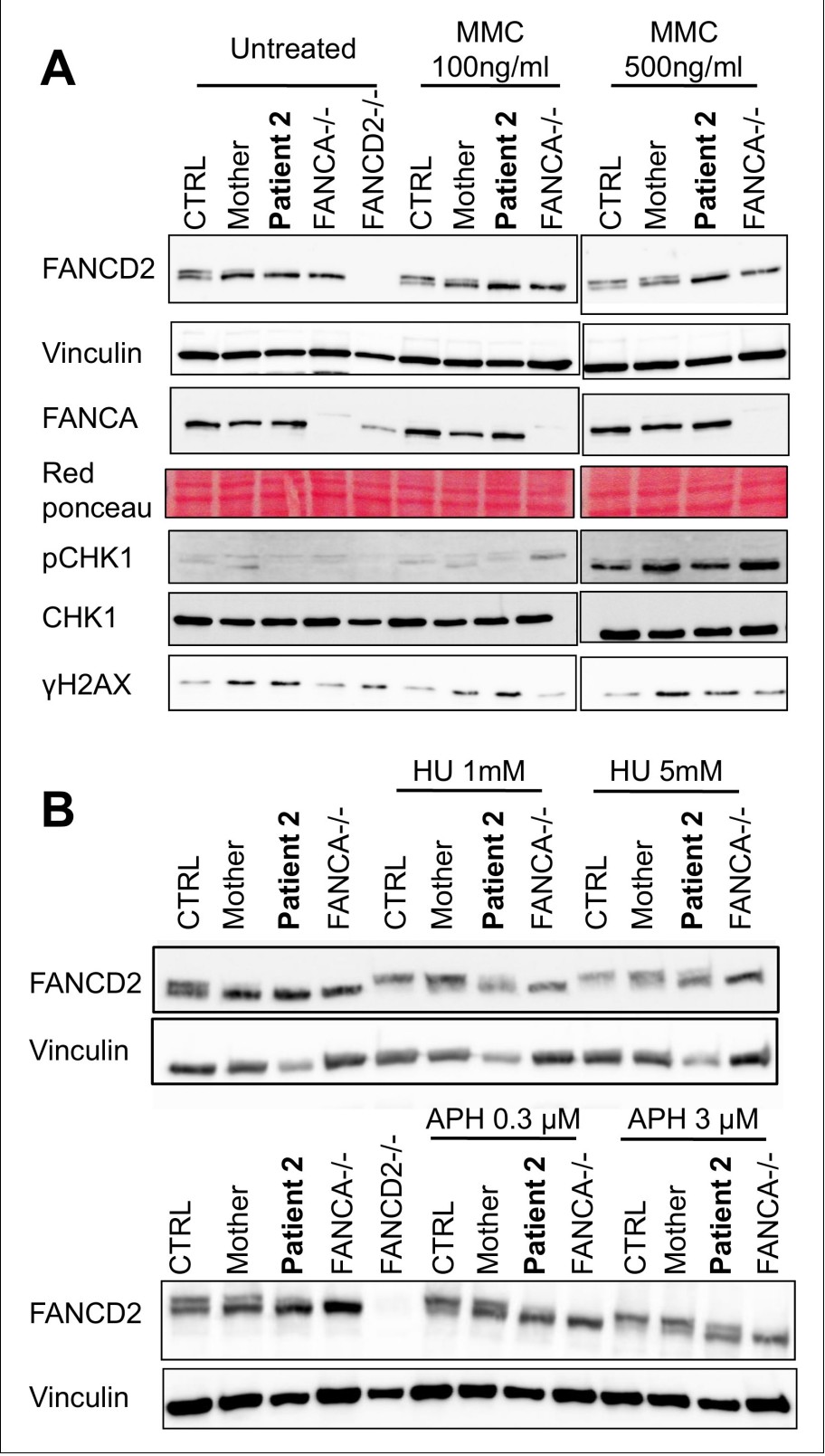

**Figure 4.** The *FANCM* 5101C>T mutation leads to altered FANCD2 monoubiquitination in response to DNA damage but not to replication inhibition. Western blot showing FANCD2 monoubiquitination, CHK1 and H2AX phosphorylation in response to MMC (**A**), or HU or APH (**B**) in cells from patients 2 and her mother. FANCA (HSC-72) and FANCD2 (GM16756) lymphoblasts were used for comparison. Each panel presents the result of a

*Figure 4 continued on next page*

*Figure 4 continued*

representative experiment on at least three independent analysis. Proteins were extracted 24 hr after exposure to genotoxins.

DOI: https://doi.org/10.7554/eLife.30490.011

manufacturer's instructions and protocols (SureSelect Human All Exon Kits Version CRE, Agilent) without modification. Libraries were sequenced on an Illumina HiSEQ 2500 as paired-end 75 bp reads. Image analysis and base calling was performed using Illumina Real Time Analysis (RTA 1.18.64) with default parameters. The various metrics of the NGS sequencing are detailed in the *Table 2*. The sequencing data was analyzed with the Illumina pipeline (CASAVA 1.8.2) for read alignment and variant calling, and an IntegraGen in-house pipeline was used for variant annotation. Annotation of kown variants was performed according to the following databases: dbSNP144 (RRID:SCR_002338), 1000 Genomes Project (release_v3.20101123) (RRID:SCR_008801), Exome Variant Server (ESP6500SI-V2-SSA137) (RRID:SCR_012761), HapMap3 (RRID:SCR_002846), ExAC r3.0 (RRID:SCR_004068), COSMIC 71 (RRID:SCR_002260), ClinVar 201507 (RRID:SCR_006169) and an Integragen internal database (containing 201 control exomes for SNPs and 130 control exomes for Indels). The variants were filtered using IntegraGen ERISv3 platform. The number of called variants and the effect of the various variant filters on the number of remaining candidate variants are shown in *Table 3*.

The sequence of the exon 20 of the FANCM gene was verified by PCR amplification and Sanger sequencing from the genomic DNA of the available individuals, using the primers FANCM-Forward (5'-AAAACTCGACGTGCAGTAATG-3') and FANCM-Reverse (5'-GAGGTTTGAAGTCTGAGACTT-3').

## Collection and processing of human ovaries

Human fetal material was provided by the Department of Obstetrics and Gynecology at the Antoine Béclère Hospital (Clamart, France) following legally induced abortions (first trimester) or therapeutic termination of pregnancy (second and third trimesters). The fetal age was evaluated by measuring the length of limbs and feet as previously described (*Evtouchenko et al., 1996*). Adult human ovaries were provided by the Laboratory of Pathology of Gustave Roussy Institute (Villejuif, France) following prophylactic removal due to breast cancer. Adult ovaries did not present any pathological aspect, showing corpora lutea at various stages of regression compatible with at least three successive ovulatory cycles. All women provided an informed consent and this study was approved by the Biomedicine Agency (reference number PFS12-002). Tissue was either snap-frozen for RNA analysis of formalin-fixed for immunohistochemistry studies.

## Expression time-course of FANCM transcripts in fetal ovaries

Human fetal material was obtained from the Antoine Béclère Hospital (Clamart, France) following legally-induced abortions or therapeutic termination of pregnancy. The fetal age was evaluated as previously described (*Evtouchenko et al., 1996*). All women provided an informed consent and this study was approved by the Biomedicine Agency (reference number PFS12-002). After collection, fetal ovaries were stored in RNA lysis buffer (RLT) (Qiagen Courtaboeuf, France) for gene expression profiling. For some experiments human gonads were dissociated and germ and somatic cells were sorted using M2A (D240/PODOPLANIN) as previously described (*Muczynski et al., 2012*). The M2A antigen was first reported as being present in testicular germ cells and germ cell tumors (*Marks et al., 1999*). It has recently been shown that the M2A antigen is also expressed in female germ cells in the human developing ovary (*Frydman et al., 2017*). Sorting was performed using a BD-Influx biohazard system (BD Biosciences; San Jose, CA; USA) with the D2-40-positive cells being mostly germ cells. Total RNA from fetal ovaries and sorted cells was extracted using the RNeasy Mini Kit (Qiagen). The 7900HT Fast Real-Time PCR System (Applied Biosystems, Foster City, CA) and SYBR-green labelling were used for quantitative RT-PCR (RT-qPCR). The comparative ΔΔcycle threshold method was used to determine the relative quantities of mRNA using ACTB (ß-actin) mRNA as reference gene for normalization. Each RNA sample was analyzed in duplicate. The sequences of oligonucleotides used for amplification are:

FANCM ForwardExp (5'-GAGGAGCTTGTCCCGCTG-3'),
ReverseExp: (5'-TGACTAGTTCTCTTACAACCTGGCAATA-3');

B-ACT Forward: (5'-TGACCCAGATCATGTTTGAGA-3'),
Reverse (5'-TACGGCCAGAGGCGTACAGG-3').

## In situ detection of FANCM protein by immunohistochemistry

Immunohistochemistry was performed on paraffin-embedded human fetal and adult ovaries as previously described (*François et al., 2017*). Briefly, human fetal ovaries were fixed in 10% neutral formalin, dehydrated, embedded in paraffin and sectioned (5 μm) as described in by (*François et al., 2017*). Slides were deparaffinized, rehydrated and heated to 98°C in 0.05% citraconic anhydride, pH 7.4 (Sigma-Aldrich Corp.), for 45 min and then blocked for 1 hr in 10% fetal calf serum at room temperature. After washing in phosphate-buffered saline, slides were incubated overnight with anti-FANCM antibody (CV5.1, Novus biologicals, Abingdon, UK, dilution 1/500) (RRID:AB_2716711). After washing, slides were incubated with goat anti-mouse secondary antibody for 90 min at room temperature and rinsed in phosphate-buffered saline. Slides were then incubated in 3,3'-diaminobenzidene (DAB substrate kit for peroxidase; Vector Laboratories, Burlingame, CA; SK-4100), and after staining development, they were counterstained with hematoxylin and mounted in Eukitt (Sigma). Similarly, for double immunostaining anti-SYCP3 (NB300-232, Novus biological) (RRID:AB_2087193) and anti-DDX4 (ab13840, Abcam) (RRID:AB_443012) antibodies were used and VIP (Vector laboratories) was used as a second substrate.

## Chromosome breakage analysis

Primary lymphocytes were cultured under standard conditions for karyotyping. Baseline and DNA damage induced chromosome breakage and rearrangement were scored blinded on 50 metaphases. DNA damage was induced with Mitomycin C (MMC, Sigma) added for 72 hr.

## Immunoblot analysis

Immunoblots were performed as described (*Bourseguin et al., 2016*). The antibodies used were: mouse monoclonal anti-FANCM antibody (CV5.1, Novus biologicals, Abingdon, UK) (RRID:AB_2716711), mouse monoclonal anti-FANCD2, (Santa-Cruz Biotechnology, Dallas, Texas, USA, SC-20022) (RRID:AB_2278211), rabbit anti-FANCA (Abcam) (Bethyl Laboratories, Montgomeryn Texas, USA, Cat# A301-980A RRID:AB_1547945), mouse anti-vinculin (Abcam) (RRID:AB_11156698), Rabbit anti-CHK1 antibody from Santa-Cruz Biotechnology (SC-8408) (RRID:AB_627257), mouse monoclonal anti-Phospho-CHK1 (Ser317) antibody from Cell Signaling Technology (#2344) (RRID:AB_331488), monoclonal anti-phospho-H2AX (Ser139) antibody from Millipore (RRID:AB_309864). To transiently deplete FANCM, HEK293 cells (RRID:CVCL_0045) were transfected with 20 nmol/L of small interfering RNA (siRNA) targeting FANCM, 5'-GGC-UAC-GUC-CAG-GAG-CGC-3' with 8 μL INTERFERin (Polyplus) in Opti-MEM. The protein bands were visualized and recorded using an ImageQuant apparatus. Western blot quantifications were performed using densitometry measures and the ImageJ software.

## Growth inhibition analysis

Measurement of MMC growth inhibition was performed as previously described (*Ridet et al., 1997*). Measurement of Mitomycin C (MMC, Sigma) growth inhibition was performed as previously described (*Bourseguin et al., 2016*). Briefly, 5−10 × 10⁵ cells/wells were seeded in 24-wells plate in 1 ml of complete culture medium, and left untreated or exposed to various concentration of MMC. After a growth period of 3–5 days, cells were counted in a Coulter counter. Growth percentage was calculated as follows: % Growth = 100 Xfinal number MMC-treated cells/final number untreated cells.

Web resources dbSNP144, https://www.ncbi.nlm.nih.gov/projects/SNP/ (RRID:SCR_002338)

1000 Genomes Project (release_v3.20101123), http://www.internationalgenome.org/ (RRID:SCR_008801)

Exome Variant Server (ESP6500SI-V2-SSA137), http://evs.gs.washington.edu/EVS/ (RRID:SCR_012761)

HapMap3, ftp://ftp.ncbi.nlm.nih.gov/hapmap/ (RRID:SCR_002846)

ExAC r3.0, http://exac.broadinstitute.org/ (RRID:SCR_004068)

COSMIC 71, http://cancer.sanger.ac.uk/cosmic (RRID:SCR_002260)

ClinVar 201507, https://www.ncbi.nlm.nih.gov/clinvar/ (RRID:SCR_006169)
Exac database: http://exac.broadinstitute.org/ (RRID:SCR_004068)
OMIM: http://omim.org/ (RRID:SCR_006437)

# Acknowledgements

The authors declare they have no conflict of interest. We thank the Universities Paris Sud, the INSERM, and the CNRS for providing supports for this study. RV and FR are respectively 'Equipe labellisée' de la Fondation pour la Recherche Medicale (grant DEQ20150331757) et de la Ligue nationale contre le cancer. We thank M. Bogliolo (Barcelona) for the gift of FANCM cDNA expressing lentiviral vectors. K.U. was supported by a fellowship from the Warsaw University of Life Sciences, Warsaw, Poland.

# Additional information

## Funding

| Funder | Grant reference number | Author |
|---|---|---|
| Fondation pour la Recherche Médicale | DEQ20150331757 | Baptiste Fouquet<br>Sandrine Caburet<br>Reiner A Veitia<br>Micheline Misrahi |
| Ligue Contre le Cancer | | Patrycja Pawlikowska<br>Filippo Rosselli |

The funders had no role in study design, data collection and interpretation, or the decision to submit the work for publication.

## Author contributions

Baptiste Fouquet, Formal analysis, Investigation, Visualization, Writing—original draft, Writing—review and editing; Patrycja Pawlikowska, Formal analysis, Investigation, Validation, Visualization; Sandrine Caburet, Formal analysis, Visualization, Writing—original draft; Celine Guigon, Laura Bellutti, Bérangère Legois, Formal analysis, Investigation; Marika Mäkinen, Laura Tanner, Marja Hietala, Bettina Bessieres, Alain Gougeon, Alexandra Benachi, Resources; Kaja Urbanska, Investigation; Gabriel Livera, Formal analysis, Investigation, Writing—original draft; Filippo Rosselli, Conceptualization, Funding acquisition, Visualization, Methodology, Writing—original draft; Reiner A Veitia, Micheline Misrahi, Conceptualization, Formal analysis, Methodology, Writing—original draft, Writing—review and editing

## Author ORCIDs

Baptiste Fouquet (iD) http://orcid.org/0000-0003-2069-1956
Sandrine Caburet (iD) http://orcid.org/0000-0002-7404-8213
Gabriel Livera (iD) http://orcid.org/0000-0001-8436-4730
Filippo Rosselli (iD) https://orcid.org/0000-0003-1080-5745
Reiner A Veitia (iD) http://orcid.org/0000-0002-4100-2681
Micheline Misrahi (iD) http://orcid.org/0000-0002-5379-8859

## Ethics

Human subjects: The study was approved by all the institutions involved. All participants gave informed consent for the study and the study was approved by the agence de Biomedecine (reference number PFS12-002).

## Decision letter and Author response

Decision letter https://doi.org/10.7554/eLife.30490.014
Author response https://doi.org/10.7554/eLife.30490.015

## Additional files

**Supplementary files**
• Transparent reporting form
DOI: https://doi.org/10.7554/eLife.30490.012

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
