## [Decision Letter]

Thank you for submitting your article "A homozygous FANCM mutation underlies familial non-syndromic primary ovarian Insufficiency" for consideration by *eLife*. Your article has been reviewed by three peer reviewers, and the evaluation has been overseen by a Reviewing Editor and Jessica Tyler as the Senior Editor. The following individuals involved in review of your submission have agreed to reveal their identity: Jordi Surralles (Reviewer #2).

The reviewers have discussed the reviews with one another and the Reviewing Editor has drafted this decision to help you prepare a revised submission.

Summary:

You describe two sisters with premature ovarian failure and homozygous FANCM mutations. Several previous studies have pointed to a role for FANCM and other FA genes in meiosis, and mouse models of FANCM deficiency also have ovarian failure. This led the authors to conclude that FANCM is causative in this family. Unfortunately, because the siblings are related and have consanguineous parents, the evidence for FANCM being the causative gene is not yet definitive. The reviewers discussed that while the genetic/molecular part of the study is technically well performed, the manuscript is limited in its methodology and findings. None-the-less, they all felt that the study is important given the general interest surrounding identification of new POI genes, and in individuals with homozygous FANCM deficiency.

Essential revisions:

The reviewers raise a number of concerns that must be adequately addressed before the paper can be accepted. Some of the required revisions will likely require further experimentation within the framework of the presented studies and techniques.

1) Genetic complementation of the FANCM deficient cells must be performed, to conclusively demonstrate the role of FANCM in the cellular phenotypes observed. As an alternative (or in addition), the authors could attempt to identify and characterize additional FANCM bialelics in a POI cohort. Either one of these analyses would exclude the possibility that undetected mutations in other FA genes (deep intronic, large deletions, mutations in low coverage/non covered regions, etc…) are present. This is important when dealing with FANCM given that the first case reported by H. Joenje and co-workers turned out to also bear mutations in FANCA.

2) Pachytene oocytes were identified as a stage of high FANCM expression. However, it is very difficult to determine meiotic stage in formalin fixed embryonic ovaries using IHC because nuclear morphology is lost. Co-staining of FANCM with meiotic markers should be performed in order to precisely determine at which stages of oogenesis FANCM is expressed. Negative (isotype) controls must also be included.

3) The authors should reference two new studies that have identified FANCM homozygous patients with cancer predisposition (Genetics in Medicine, 2017) as well as a previous study that links primary ovarian failure to other breast cancer genes such as BRCA1: Oktay et al., 2010.

4) A conclusion is made that the patient cells are not sensitive to HU and APH, but this is based only on an absence of FANCD2 ubiquitination. A sensitivity assay to one or both of these drugs must be included, or this point should be withdrawn.

[Editors' note: further revisions were requested prior to acceptance, as described below.]

Thank you for resubmitting your work entitled "A homozygous *FANCM* mutation underlies a familial case of non-syndromic primary ovarian Insufficiency" for further consideration at *eLife*. Your revised article has been favorably evaluated by Jessica Tyler (Senior editor) and the Reviewing editor.

The manuscript has been improved but there are some minor remaining issues that need to be addressed before acceptance, as outlined below:

1) You have not correctly cited the recent FANCM homozygotes papers. In particular, in the Introduction "The only phenotype associated to FANCM biallelic mutations thus far is a predisposition to early-onset breast cancer" and in the Discussion section but Bogliolo et al. describe only *male* patients, who develop B-LL and head and neck cancers. Both papers also highlight a chemotherapy sensitivity in FANCM homozygotes, which should be considered a "phenotype". A more appropriate sentence might be: "The only phenotypes associated to FANCM biallelic mutations thus far are cancer predisposition, in particular early-onset breast cancer in females, and chemosensitivity". In addition, it is of significant relevance that 3/5 female FANCM homozygotes in the Catucci et al. paper are characterised as having "early onset menopause", which is indicative of POI and thus supportive of the findings in your paper. These points need to be added to the manuscript.

2) In Frydman, et al. paper, the antibody used seems to be expressed in oogonia and not meiotic oocytes. This means that while you have likely sorted germ cells from somatic cells, you probably do not have the population that has the highest FANCM protein expression and or the germ cells that are actually undergoing recombination (this might explain why you didn't see a statistically significant increase in mRNA expression in Figure 2). You should clearly specify in what stage of "germ cells" you are measuring expression.

3) In subsection “FANCM is expressed in germ cells of human fetal and adult ovaries” you refer to "…an increase in ovaries containing germ cells progressing into meiotic prophase 1". The n=2 at 14-17wks prevents statistical analysis, so it is not appropriate to refer to an increase in this instance.

---

## [Author Response]

Essential revisions:1) Genetic complementation of the FANCM deficient cells must be performed, to conclusively demonstrate the role of FANCM in the cellular phenotypes observed. As an alternative (or in addition), the authors could attempt to identify and characterize additional FANCM bialelics in a POI cohort. Either one of these analyses would exclude the possibility that undetected mutations in other FA genes (deep intronic, large deletions, mutations in low coverage/non covered regions, etc…) are present. This is important when dealing with FANCM given that the first case reported by H. Joenje and co-workers turned out to also bear mutations in FANCA.

We have exhaustively looked for the presence of variants in a list of FA genes (new Table 4). The very good coverage of each gene (80-90X in the POI patients) excludes the possibility of having missed a variant due to a lack of coverage. The presence of heterozygous variants, both in coding regions and elsewhere (UTRs, the covered intronic sequences, etc.), in one or both POI patients, excludes the possibility of hemizygosity due to heterozygous deletions. The ratios of the number of reads of the alleles for these heterozygous variants were unbiased (i.e., close to 1) and argue against gene deletions. The detection of variants outside the coding portions of the exons also argues against the possibility of partial deletions. Exome-based CNV analysis showed that the detected CNVs were unrelated to the disease as they were either not common to the two sisters or were present in the mother. Finally, the total lack of correctly segregating variants (i.e common to both POI patients and heterozygous in the mother) in FA genes excludes the possibility of their involvement in the pathology. These analyses show that the cellular phenotypes observed in the cells of the patients are not due to mutations in FA genes other than FANCM. These results have been added to subsection “Identification of a FANCM mutation by whole-exome sequencing and molecular studies.” and in Table 4.

Moreover, we performed a genetic complementation of the FANCM deficient cells of patient 2 wich significantly increased their resistance to MMC exposure (with p values <0.05 or <0.01 according to the concentration of MMC used) and restored FANCD2-monoUbiquitination. The results are shown in subsection “The *FANCM* mutation leads to hypersensitivity to Mitomycin C and altered FANCD2 monoubiquitination in response to DNA interstrand crosslinks but not to replication inhibition” and in Figure 3 and E. See also subsection “Cell lines and infection with a lentiviral vector” and subsection “Whole-exome and Sanger sequencing”.

2) Pachytene oocytes were identified as a stage of high FANCM expression. However, it is very difficult to determine meiotic stage in formalin fixed embryonic ovaries using IHC because nuclear morphology is lost. Co-staining of FANCM with meiotic markers should be performed in order to precisely determine at which stages of oogenesis FANCM is expressed. Negative (isotype) controls must also be included.

The identification of meiotic stages was performed in histological pieces that have we previously characterized (Frydman, 2017) and based on the chromatin features. Unfortunately, due to incompatibility of the antibodies in double immunostaining experiments it was not possible to obtain satisfying results with the SYCP3/FANCM co-staining after trying several IF protocols. We thus performed immunohistochemistry using VIP, a washable substrate, to stain first FANCM and subsequently SYCP3. We now also provide evidence that FANCM is present in DDX4+ cells (germ cells). Additionally, we provide a negative control. See subsection “In situ detection of FANCM protein by immunohistochemistry”, subsection “FANCM is expressed in germ cells of human fetal and adult ovaries**”** and new Figure 2.

3) The authors should reference two new studies that have identified FANCM homozygous patients with cancer predisposition (Genetics in Medicine, 2017) as well as a previous study that links primary ovarian failure to other breast cancer genes such as BRCA1: Oktay et al., 2010.

These references have been added to the Introduction, the Discussion section and to the Reference list.

4) A conclusion is made that the patient cells are not sensitive to HU and APH, but this is based only on an absence of FANCD2 ubiquitination. A sensitivity assay to one or both of these drugs must be included, or this point should be withdrawn.

Accordingly, to the suggestion, the point has been withdrawn.

[Editors' note: further revisions were requested prior to acceptance, as described below.]

1) You have not correctly cited the recent FANCM homozygotes papers. In particular, in the Introduction "The only phenotype associated to FANCM biallelic mutations thus far is a predisposition to early-onset breast cancer" and in the Discussion section but Bogliolo et al. describe only *male* patients, who develop B-LL and head and neck cancers. Both papers also highlight a chemotherapy sensitivity in FANCM homozygotes, which should be considered a "phenotype". A more appropriate sentence might be: "The only phenotypes associated to FANCM biallelic mutations thus far are cancer predisposition, in particular early-onset breast cancer in females, and chemosensitivity". In addition, it is of significant relevance that 3/5 female FANCM homozygotes in the Catucci et al. paper are characterised as having "early onset menopause", which is indicative of POI and thus supportive of the findings in your paper. These points need to be added to the manuscript.

According to the suggestion of the reviewer we have changed the sentence in the Introduction and the Discussion section: “The phenotypes associated to FANCM biallelic mutations thus far are cancer predisposition, in particular early-onset breast cancer in females, and chemosensitivity", and also in the Discussion section: “In one study (Catucci et al., 2017) two female patients out of five had premature menopause and one patient had an early reduction of the ovarian reserve according to her AMH levels”.

2) In Frydman, et al. paper, the antibody used seems to be expressed in oogonia and not meiotic oocytes. This means that while you have likely sorted germ cells from somatic cells, you probably do not have the population that has the highest FANCM protein expression and or the germ cells that are actually undergoing recombination (this might explain why you didn't see a statistically significant increase in mRNA expression in Figure 2). You should clearly specify in what stage of "germ cells" you are measuring expression.

This is correct; M2A or D2-40 immunolabelling is strongly present in oogonia and sharply decreases from meiotic entry onwards. Our unpublished data indicate the expression of meiotic genes in some D2-40-positive cells. Nonetheless, it is likely that the majority of sorted cells are oogonia and not cells at pachytene stage, we thus changed the text in subsection “FANCM is expressed in germ cells of human fetal and adult ovaries” to specify this: “Cell-sorting experiments conducted in 8-12 wpf ovaries indicated that FANCM transcripts were predominant in oogonial cells (D2-40-positive) compared to somatic cells (Figure 2).”

3) In subsection “FANCM is expressed in germ cells of human fetal and adult ovaries” you refer to "…an increase in ovaries containing germ cells progressing into meiotic prophase 1". The n=2 at 14-17wks prevents statistical analysis, so it is not appropriate to refer to an increase in this instance.

Text has been changed accordingly in subsection “FANCM is expressed in germ cells of human fetal and adult ovaries”: “qRT-PCR in human fetal ovaries demonstrated that FANCM mRNAs were expressed throughout ovarian development (5-32 weeks post-fertilization, wpf) (Figure 2). Of note, expression tended to be higher than average in 14 and 17 wpf ovaries that are stages containing the highest proportion of germ cells progressing into meiotic prophase I.”